# Comparative Performance of Recombinant GRA6, GRA7, and GRA14 for the Serodetection of *T. gondii* Infection and Analysis of IgG Subclasses in Human Sera from the Philippines

**DOI:** 10.3390/pathogens11020277

**Published:** 2022-02-21

**Authors:** Rochelle Haidee Ybañez, Yoshifumi Nishikawa

**Affiliations:** 1National Research Center for Protozoan Diseases, Obihiro University of Agriculture and Veterinary Medicine, Obihiro 080-8555, Japan; rochelledybanez@gmail.com; 2Institute of Molecular Parasitology and Protozoan Diseases, Main Campus and College of Veterinary Medicine, Barili Campus, Cebu Technological University, Cebu City 6000, Philippines

**Keywords:** ELISA, GRA6, GRA7, GRA14, human, IgG subclass, *Toxoplasma gondii*

## Abstract

Highly specific and sensitive diagnostic methods are vital for the effective control and treatment of toxoplasmosis. Routine diagnosis is primarily serological because *T. gondii* infections stimulate persistently high IgG antibody responses. The sensitivity and specificity of methods are crucial factors for the proper diagnosis of toxoplasmosis, primarily dependent on the antigens used in different assays. In the present study, we compared the serodiagnostic performances of three recombinant dense granule antigens, namely, the GRA6, GRA7, and GRA14, to detect IgG antibodies against *T. gondii* in human sera from the Philippines. Moreover, we evaluated the IgG1, IgG2, IgG3, and IgG4 responses against the different recombinant antigens, which has not been performed previously. Our results revealed that the TgGRA7 has consistently displayed superior diagnostic capability, while TgGRA6 can be a satisfactory alternative antigen among the GRA proteins. Furthermore, IgG1 is the predominant subclass stimulated by the different recombinant antigens. This study’s results provide options to researchers and manufacturers to choose recombinant antigens suitable for their purpose.

## 1. Introduction

Toxoplasmosis is known to infect nearly one-third of the world’s human population [1]. The accurate detection of *T. gondii* infection through highly specific and sensitive diagnostic methods is vital for the effective control and treatment of toxoplasmosis [2,3]. *T. gondii* infections in humans are characterized by persistent high IgG antibody titers [4,5]. Thus, routine diagnosis for toxoplasmosis is mainly through detecting specific *T. gondii* antibodies.

The sensitivity and specificity of methods are crucial factors for the proper diagnosis of toxoplasmosis to avoid false-positive and false-negative results. These factors are primarily dependent on the antigens used in different assays. The recombinant antigens have been considered alternative diagnostic markers to replace the native antigens, given the tedious standardization and high cost to produce crude *T. gondii* antigens. Aside from improving *T. gondii* diagnosis, differentiating the different phases of infection using recombinant antigens has also been widely recognized [6,7]. Among the recombinant antigens that showed outstanding serodiagnostic performance are the SAG1 [8,9,10] and SAG2 [11,12] of the surface antigen (SAG) family; the dense granule (GRA) proteins GRA1 [8], GRA2 [13,14], GRA 3 [9], GRA5 [15], GRA6 [16,17], GRA7 [10,18,19,20], and GRA8 [21,22]; the rhoptry proteins ROP1 [14,23], ROP2 [24], and ROP8 [25]; and MAG1 [26] and MIC2 [9].

Moreover, several studies have investigated the dynamics of human IgG subclasses in their varied responses to different infectious agents. The prevalent IgG responses to protein antigens are IgG1 and IgG3, while IgG2 is against carbohydrates. IgG1 and IgG3 are stimulated for viral antigens, while bacterial antigens primarily elicit the IgG2 subclass [27,28]. These IgG responses also vary depending on the host organism [29].

In the present study, we compared the serodiagnostic performances of three recombinant GRA antigens, namely, the GRA6, GRA7, and GRA14, to detect IgG antibodies against *T. gondii* in human sera from the Philippines. Furthermore, we evaluated the IgG1, IgG2, IgG3, and IgG4 responses against the recombinant antigens, which has not been conducted previously. Our findings confirmed the superior diagnostic potential of TgGRA7 and the suitability of the TgGRA6 as an alternative antigen for toxoplasmosis serodiagnosis. In addition, the first use of TgGRA14 as a diagnostic marker for human toxoplasmosis is documented in this study. Furthermore, IgG1 is the predominant subclass recognized by the different recombinant antigens.

## 2. Results

In the present study, we assessed the reactivity of IgG and IgG subclasses (IgG1, IgG2, IgG3, IgG4) in human sera from the Philippines against three recombinant antigens expressed in E. coli as GST-fused (TgGRA7 and TgGRA14) and His-tagged (TgGRA6) proteins through indirect ELISA. We also compared the performance of the recombinant antigens with Platelia IgG-ELISA (com-ELISA). The ELISAs using TgGRA6 (25 positives of 88 samples) and TgGRA7 (27 positives of 88 samples) obtained similar results as the com-ELISA (27 positives of 88 samples) for the detection of IgG antibodies (Table 1). The TgGRA14 showed the lowest IgG detection rate (11 positives of 88 samples) (Table 1). The TgGRA7-ELISA detection results were in perfect agreement with the com-ELISA. For the TgGRA6- and TgGRA14-ELISA, 2 and 16 human serum samples were found below the cut-off values, respectively. These samples were judged positive by com-ELISA (Figure 1). Similarly, the TgGRA14-ELISA showed the lowest sensitivity (40.7%), although with 100% specificity against the com-ELISA. Meanwhile, high sensitivity (92.6–100%), specificity (100%), and kappa values (0.945–1) in the ELISAs using TgGRA6 and TgGRA7 were found (Table 2).

The 27 serum samples that detected IgG antibodies using any of the three recombinant antigens were further evaluated for the reactivity of specific IgG subclasses by ELISA (Figure 2). TgGRA6 and TgGRA7 showed significant differences (*p*-value < 0.001) in the detection results of anti-*Toxoplasma* IgG subclasses from human sera (Table 3). Moreover, the detection rate of IgG1 in both antigens was significantly higher than IgG2, IgG3, and IgG4 (Table 3).

## 3. Discussion

Our results revealed that IgG antibodies reacted more strongly and frequently with TgGRA6 and TgGRA7 ELISAs than TgGRA14. Detection rates for both antigens were similar to results obtained using the com-ELISA. It shows the importance of recombinant antigens in the serological detection of *T. gondii* infections in humans, as isolating *T. gondii* is difficult [30]. Many recombinant antigens produced in *E. coli* have been examined and confirmed to be reliable markers for *T. gondii* infection diagnosis [8,10,11,24]. Regardless of the clinical presentations in the infected individuals, *T. gondii* infections elicit intense and often lasting humoral immune responses characterized by high antibody titers [4,5]. Thus, serological assays to detect specific antibodies are still the favored method for diagnosing acute or chronic infections of *T. gondii* [31].

In the present study, the high sensitivity and specificity of TgGRA6-ELISA and its perfect agreement with conventional ELISA test reveal that TgGRA6 is a promising alternative antigen for toxoplasmosis serodiagnosis in the Philippines. The usefulness of TgGRA6 as a serodiagnostic antigen for human infections was previously validated in several ELISAs in Poland [11], Germany [16], Iran [17], and France [32]. A rTgGRA6 demonstrated high reactivity with specific IgG-positive human sera, reaching a sensitivity of 96% in an IgG ELISA test [32]. An ELISA using a GST-tagged TgGRA6 obtained an 89% sensitivity and 99.6% specificity for detecting IgG [16]. The same study also reported that the recombinant antigen successfully differentiated acute from chronic infections, with a sensitivity of 86% [16]. Another ELISA utilizing a rTgGRA6 with His-tag domain was able to detect more recent toxoplasmosis (93.9%) than chronic cases (60.6%) [11]. Furthermore, an IgM-ELISA using TgGRA6 demonstrated a high sensitivity of 97.1% in discriminating acute from past *Toxoplasma* infections in pregnant women, which confirms a remarkable correlation with VIDAS Toxo IgM kit [17].

An immunochromatographic test using TgGRA7 revealed a 93.1–100% sensitivity and 100% specificity for detecting IgG and IgM and/or IgG antibodies [20]. This affirms TgGRA7’s outstanding antigenic potential vis-à-vis standard tests. The GRA proteins are among the antigens well recognized for their excellent diagnostic ability [14,18,22]. Their vast immunogenic properties [33,34] stimulate cell-mediated or antibody-dependent immunity [35]. Moreover, the TgGRA7 is an excellent serodiagnostic marker for human toxoplasmosis, with a sensitivity of 81–98.9% and specificity of 98-100% using ELISA [9,10,18,36]. Furthermore, this antigen is more correlated with acute toxoplasmosis, with reported sensitivity ranging from 75 to 100% in ELISA [8,10,18,24].

Although the TgGRA14-ELISA in our current study demonstrated high specificity, it had low sensitivity to detect IgG antibodies in human sera. The current findings corroborated with a previous study where a TgGRA14-ELISA revealed low reactivity using experimentally infected mice characterized by lower OD values compared to TgGRA7. It also registered a sensitivity of only 81.25% for detecting IgG and/or IgM antibodies to infection in field pig sera, which was lower than TgGRA7 (90.63%) [37].

Apart from the present study, TgGRA14 has not been utilized yet as an ELISA antigen for human toxoplasmosis serodiagnosis. The GRA14, along with other GRA proteins, is found on the parasitophorous vacuole (PV) membrane extensions linking neighboring PVs [38]. It is also expressed in all life stages of *T. gondii*, which contributes to its robust antigenic properties [39]. Previous studies using DNA immunizations with GRA14 enhanced cellular and humoral responses, leading to relative protection against toxoplasmosis in mice [40,41]. However, the sensitivities and specificities of serological assays using recombinant diagnostic antigens may vary. These differences may be due to variances in cloning approaches, protocols used during the purification of recombinant proteins, and standards used in the data analyses [6,42]. Moreover, recombinant antigens often lose their antigenicity due to incorrect folding and probable contamination with *E. coli* antigens during protein expression [43,44,45,46].

For the IgG subclasses, our results found the IgG1 to be the primary subclass response stimulated by all the examined recombinant antigens, similar to previous studies showing the predominance of IgG1 antibodies in the humoral immune response to *T. gondii* infections in humans [29,47,48,49]. IgG3 and IgG4 antibodies were also detected in this study, although less frequently than the IG1, while IgG2 antibody detection was zero to very minimal. These results were similar to the ELISA study of [48] using sera from patients clinically suspected of toxoplasmosis. Furthermore, lesser responses in the IgG2 and IgG3 subclasses were observed among human sera from individuals with acute and chronic *T. gondii* infections [29]. Contrastingly, an elevated IgG2 response in a standardized ELISA using a cyst antigen was noted in the CSF of patients with cerebral toxoplasmosis [49].

The current and previous studies may explain the preferential recognition of *T. gondii* antigens by lgG antibodies where lgG1; IgG3; and, to a lesser extent, IgG4 react with protein antigens [50], while IgG2 reacts to polysaccharide antigens [51,52]. The high IgG2 levels detected in [49] could be explained by the presence of polysaccharides in the *T. gondii* cysts used as antigen in the ELISA [52]. Large quantities of polysaccharides are produced and stored in the bradyzoite cytoplasm and cyst wall during cyst formation [53,54]. Moreover, the predominance of IgG1 at different infection stages in immunocompetent individuals is presumed to be related to a T-cell control of humoral response during toxoplasmosis [47]. Some studies also suggest that lgG1 and lgG3 stimulate phagocytosis of parasites by stimulating the binding of mononuclear cells to the parasite [55] and activating the complement system [56]. Furthermore, other studies proposed that IgG subclasses specific to *T. gondii* could be markers of clinical outcome of toxoplasmosis [57,58,59].

In an ELISA using tachyzoite antigens, elevated levels in IgG1, IgG2, IgG3, and/or IgG4 in mothers and/or newborns were associated with offspring clinical problems or vertical transmission [57]. The association between clinical signs of congenital toxoplasmosis and IgG subclasses using antirMIC3-ELISA revealed that the IgG2 or IgG4 reactivity was linked to the occurrence of retinochoroidal lesions and intracranial calcifications [58]. Meanwhile, detecting *T. gondii*-specific IgG4 antibodies in serum and/or CSF samples reinforced the diagnosis of cerebral toxoplasmosis in HIV-infected patients [59].

## 4. Materials and Methods

### 4.1. Human Serum Samples

In this study, we tested 88 human serum samples from the Philippines, which a licensed phlebotomist aseptically collected, separated by centrifugation, and stored at −20 °C until further use.

### 4.2. Production of GST-Fused Recombinant TgGRA7 and TgGRA14

All the DNA samples used as a template for the amplification of the coding regions of the target antigens were from the type I RH strain of *T. gondii* tachyzoites. Previously described protocols were adopted for the amplification and cloning of the TgGRA7 [20,37] and TgGRA14 [37] into the *Escherichia coli* expression vector pGEX-4T1 (G.E. Healthcare, Amersham, Buckinghamshire, UK). The recombinant proteins of TgGRA7 and TgGRA14 were eventually transformed in *E. coli* BL21 for protein expression.

The transformed plasmids were grown in L.B. media supplemented with 50 µg/mL of ampicillin at 37 °C until an optical density (O.D.) at 600 nm of 0.6 was reached. Protein production was induced by isopropyl-b-D-thiogalactopyranoside (IPTG) to a final concentration of 1 mM. The recombinant proteins of TgGRA7 (rTgGRA7) and TgGRA14 (rTgGRA14) were expressed as GST fusion proteins in the *E. coli* (Takara Bio, Inc., Shiga, Japan). According to the manufacturer’s instructions, the GST tags of the rTGRA7 and rTgGRA14 were removed with thrombin protease (G.E. Healthcare). SDS-PAGE results revealed that the rTgGRA7 and rTgGRA14 were 29 kDa and 30 kDa proteins, respectively. Protein concentrations were quantified using a bicinchoninic acid (BCA) protein assay kit (Thermo Fisher Scientific, Inc., Rockford, IL, USA) before storage at −30 °C until use.

### 4.3. Production of His-Tagged Recombinant TgGRA6

The PrimeScript™ II 1st strand cDNA Synthesis Kit (Takara Bio, Inc., Shiga, Japan) was used to obtain the cDNA template for the PCR amplification of the TgGRA6. The following primers containing BamHI and HindIII recognition sequences (bold): 5′-AAGGATCCATGGCAGCAGACAGCGGTGGT-3′ (forward) and 5′-CGAAGCTTTCTGTGGCGTTTCTGTGTTCG-3′ (reverse) were used to facilitate cloning. The target amplicon (345 bp) was digested using the same set of restriction enzymes and ligated into the pET30b vector in-frame of the His-tag domain at the C-terminal of the fusion protein. The correct sequence of the His-tagged protein was also confirmed via double-enzyme digestion and sequencing.

The recombinant vector was transformed in BL21 (DE3) *E. coli* and cultivated in L.B. media supplemented with 100 µg/mL of kanamycin at 37 °C until an optical density (O.D.) at 600 nm of 0.6 was reached. Induction with 1 mM IPTG was performed 16 h at 25 °C with vigorous shaking (160 rpm). The culture was centrifuged to harvest the cells. The pellet was resuspended in lysis buffer (100 mM NaH_2_PO_4_, 10 mM Tris–HCl, 8 M urea; pH 8.0) with lysozyme (1 mg/mL), mixed gently using a rotary shaker for 1 h, and disrupted by sonication (ten 10 s cycles at 55% power). Centrifugation at 10,000 rpm for 30 min at 4 °C was performed to remove insoluble debris. The supernatant was added into 50% Ni-NTA beads (Qiagen, Germantown, MD, USA) previously stabilized in lysis buffer and mixed via mild rotation at 4 °C for 1 h. The mixture was then loaded into the Ni-NTA poly-prep chromatography columns (Bio-Rad, Hercules, CA, USA). The unbound proteins were removed by washing buffer (100 mM NaH_2_PO_4_, 10 mM Tris-Cl, 8 M urea; pH 6.3), which was performed twice. The TgGRA6 protein was eluted 3–5 times from the column using elution buffer (100 mM NaH_2_PO_4_, 10 mM Tris-Cl, 8 M urea; pH 5.9). The recombinant His-tagged TgGRA6 (rTgGRA6) was a 13 kDa protein, as confirmed using SDS-PAGE analysis.

The eluted rTgGRA6 fractions were pooled and dialyzed in reducing urea concentrations (6 M, 4 M, 2 M, 1 M) for 6 h each. Final dialysis was against PBS, twice for 6 h each. A BCA protein assay kit (Thermo Fisher Scientific, Inc., Carlsbad, CA, USA) was also used to determine the protein concentration before storage at −30 °C until further use.

### 4.4. Indirect ELISA Using a Commercial Kit

Performance of the ELISA using the Platelia Toxo IgG (Bio-Rad, Hercules, CA, USA) followed previously documented protocols [20]. Interpretation of results was according to the manufacturer’s instructions.

### 4.5. Indirect ELISA Using TgGRA6, TgGRA7, and TgGRA14

The rTgGRA6, rTgGRA7, and rTgGRA14 were diluted in 50 mM carbonate-bicarbonate buffer (pH 9.6) at a final concentration of 0.1 μM. Then, each well of the ELISA plate (Nunc, Roskilde, Denmark) was coated with 50 μL of the diluted antigen and incubated overnight at 4 °C. The succeeding steps were performed as previously described [20]. For the detection of IgG subclasses, the same procedure was followed as above, except for the fact that secondary antibodies used were horseradish peroxidase (HRP)-conjugated mouse anti-human IgG1, IgG2, IgG3, or IgG4 (Thermo Fisher Scientific, USA) diluted at 1:1000.

The cutoff points for all antigens were determined as the mean value at OD415nm plus three standard deviations of human sera (N = 10) previously confirmed negative using commercial ELISA (Platelia Toxo IgG; Bio-Rad, Hercules, CA, USA) and LAT (Toxocheck-MT; Eiken Chemical, Tokyo, Japan) kits.

### 4.6. Statistical Analyses

The ELISA results were encoded to Microsoft Excel using appropriate coding for statistical analyses. Descriptive statistics were employed. The statistical significance of the test results was evaluated using analysis of variance with post hoc analyses. *p*-values of less than 0.05 were considered significant. An online statistical tool was used to calculate the sensitivity, specificity, and kappa values with a 95% confidence interval (http://vassarstats.net/ accessed on 9 June 2021). The strength of agreement was graded with kappa values of fair (0.21 to 0.40), moderate (0.41 to 0.60), substantial (0.61 to 0.80), and very good (0.81 to 1.00). Graphs were constructed using GraphPad Prism 9 (GraphPad Software Inc., San Diego, CA, USA).

## 5. Conclusions

All three recombinant proteins developed were found to detect IgG antibodies in human sera from the Philippines, albeit with varied performances. The TgGRA7 has consistently displayed superior diagnostic capability, while TgGRA6 could be a suitable alternative antigen among the GRA proteins. Moreover, IgG1 is the predominant subclass stimulated by the different recombinant antigens. Therefore, this study’s results provide options to researchers and manufacturers to choose recombinant antigens suitable for their purpose. Furthermore, analysis of the IgG subclass antibody response to *T. gondii* could be a potential tool to understand better the pathogenesis, diagnosis, and probable approach for treating toxoplasmosis.

## Figures and Tables

**Figure 1 pathogens-11-00277-f001:**
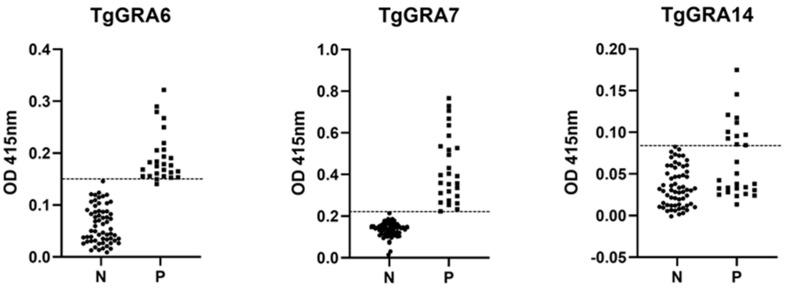
Detection of IgG antibodies against TgGRA6, TgGRA7, and TgGRA14 by indirect ELISA. Each symbol represents the mean values of the duplicate wells for each serum sample. The round and square symbols represent the negative (N) and positive (P) samples determined by Platelia IgG-ELISA, respectively. Antibody titers in OD (optical density at 415 nm) are plotted on the y-axis. Horizontal broken lines are the cut-off values for each recombinant antigen: 0.1506 (TgGRA6), 0.2205 (TgGRA7), and 0.0800 (TgGRA14).

**Figure 2 pathogens-11-00277-f002:**
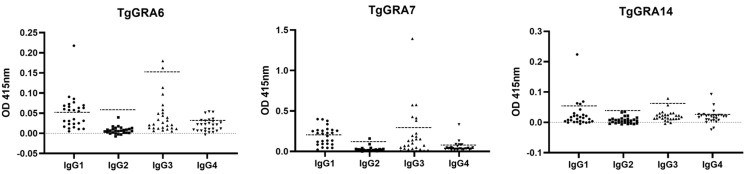
IgG subclass responses to TgGRA6, TgGRA7, and TgGRA14 by indirect ELISA (N = 27). The scatter plot shows mean OD values at 415 nm. Horizontal broken lines are the cut-off values for each recombinant antigen: TgGRA6 (IgG1, 0.0533; IgG2, 0.062; IgG3, 0.1517; IgG4, 0.0325), TgGRA7 (IgG1, 0.2162; IgG2, 0.1419; IgG3, 0.3246; IgG4, 0.0772), and TgGRA14 (IgG1, 0.0583; IgG2, 0.0275; IgG3, 0.0064; IgG4, 0.0261).

**Table 1 pathogens-11-00277-t001:** Comparison of IgG detection by ELISA using recombinant TgGRA6, TgGRA7, and TgGRA14 in human sera with commercial ELISA as reference test (N = 88).

com-ELISA ^a^	TgGRA6		TgGRA7		TgGRA14	
	(−)	(+)	(−)	(+)	(−)	(+)
Negative (−)	61	0	61	0	61	0
Positive (+)	2	25	0	27	16	11
Total	63	25	61	27	77	11

^a^ Commercial ELISA: Platelia™ Toxo IgM/IgG (Bio-Rad, Hercules, CA, USA).

**Table 2 pathogens-11-00277-t002:** Sensitivity and specificity of different ELISAs using recombinant proteins to detect specific *T. gondii* IgG antibodies using a commercial ELISA as a reference test.

Parameters	TgGRA6	TgGRA7	TgGRA14
Sensitivity (%)	92.6	100	40.7
Specificity (%)	100	100	100
Kappa value	0.945	1	0.488

**Table 3 pathogens-11-00277-t003:** Detection of *anti-Toxoplasma* IgG subclasses from human sera using TgRA6, TgGRA14, and TgGRA7 (N = 27).

Recombinant Antigens	IgG1	IgG2	IgG3	IgG4	*p*-Value
	No. (%)	No. (%)	No. (%)	No. (%)	
TgGRA6	14 (51.90)	0 (0)	2 (7.40)	6 (22.22)	< 0.001 *^,a,b,c^
TgGRA7	14 (51.85)	1 (3.70)	6 (22.22)	6 (22.22)	< 0.001 *^,d,e,f^
TgGRA14	4 (14.81)	0 (0)	1 (3.70)	5 (18.52)	0.057

* *p* < 0.05 is considered statistically significant (ANOVA); ^a^ TgGRA6-IgG1 vs. TgGRA6-IgG2; ^b^ TgGRA6-IgG1 vs. TgGRA6-IgG3; ^c^ TgGRA6-IgG1 vs. TgGRA6-IgG4; ^d^ TgGRA7-IgG1 vs. TgGRA7-IgG2; ^e^ TgGRA7-IgG1 vs. TgGRA7-IgG3; ^f^ TgGRA7-IgG1 vs. TgGRA7-IgG4

## Data Availability

The data presented in this study are available on request from the corresponding author.

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
