# Peer review of "Comparative Performance of Recombinant GRA6, GRA7, and GRA14 for the Serodetection of T. gondii Infection and Analysis of IgG Subclasses in Human Sera from the Philippines"

_pathogens, 2022, doi:10.3390/pathogens11020277_

Round 1
Reviewer 1 Report
The paper entitled “Comparative performance of recombinant GRA6, GRA7, and 3 GRA14 for the serodetection of T. gondii infection and analysis 4 of IgG subclasses in human sera from the Philippines” by Ybañez and Nishikawa describes the performance of rTgGRA6, rTgGRA7, and rTgGRA14, to detect total IgG, IgG1, IgG2, IgG3 and IgG4 antibodies against T. gondii in human sera from the Philippines.
The text is clearly written and easy to read.
Major comments
In the materials and methods section there is no reference to ethical issues related to the collection of blood samples in humans.
Minor comments
ABSTRACT
Lines 25-26: I have some doubts about the pertinence of the last sentence of the abstract, since the study does not evaluate IgG subclasses as a tool to study pathogenesis, diagnosis, or treating protocols. The study only proves that recombinant GRA antigens can be used to detect different subclasses of IgG.
INTRODUCTION
Line 60: Please consider replace the word “stimulated” by recognized, identified, etc.
MATERIAL AND METHODS
Lines 217-219: BamHI and HindIII recognition sequences do not appear to be highlighted in bold.
Line 248: “The recombinant TgGRA6,…” Please use the abbreviation created on line 207.
RESULTS
Line 68: I assume the abbreviation com-ELISA was not created. You can do this on line 67 “Platelia IgG-ELISA”
Lines 89-92: I think this paragraph would look better in the material and methods section. From the observation of Tsable 2 it is assumed that "The strength of agreement" was very good/perfect for rTGGRA6 and 7, and moderate for rTgGRA14. However, these results should be presented in the Results section in writing.
Question: The 27 rTgGRA7 positive sera were tested for the different immunoglobulin subclasses using the 3 recombinant antigens. I don't understand why 27 sera were tested for rTgGRA14 and rTgGRA6, when only 11 and 25 sera were positive for total IgG with these recombinant proteins.
Question: How was the cut-off calculated for the IgG subclasses? Does the last paragraph in section 4.5 apply to detection of total IgG and IgG subclasses?
DISCUSSION
Lines 135 and 153: What means ICT and PVs?
Author Response
Major comments
In the materials and methods section there is no reference to ethical issues related to the collection of blood samples in humans.
Response: There is a specific section at the end of the paper, the Institutional Review Board Statement in lines 284-287 of the revised paper (Lines 287-290 of the original paper) for the ethical approval obtained prior the blood collection. Thus, no reference is added anymore in the Materials and Methods section.
Minor comments
ABSTRACT
Lines 25-26: I have some doubts about the pertinence of the last sentence of the abstract, since the study does not evaluate IgG subclasses as a tool to study pathogenesis, diagnosis, or treating protocols. The study only proves that recombinant GRA antigens can be used to detect different subclasses of IgG.
Response: We deleted the statement “Analysis of the IgG subclass antibody response to T. gondii could be a potential tool to understand better the pathogenesis, diagnosis, and probable approach for treating toxoplasmosis” in the revised manuscript.
INTRODUCTION
Line 60: Please consider replace the word “stimulated” by recognized, identified, etc.
Response: Thank you for the suggestion. The word “stimulated” is now replaced by “recognized”.
MATERIAL AND METHODS
Lines 217-219: BamHI and HindIII recognition sequences do not appear to be highlighted in bold.
Response: Sequences of the restriction enzymes BamHI (GGATCC) and HindIII (AAGCTT) were highlighted in bold in line 215-216.
Line 248: “The recombinant TgGRA6,…” Please use the abbreviation created on line 207.
Response: Revised into. “The rTgGRA6, rTgGRA7, and rTgGRA14 ...” in line 245 of the revised paper.
RESULTS
Line 68: I assume the abbreviation com-ELISA was not created. You can do this on line 67
“Platelia IgG-ELISA”
Response: Platelia IgG-ELISA (com-ELISA) abbreviation added to the sentence in line 67.
Lines 89-92: I think this paragraph would look better in the material and methods section. From the observation of Table 2 it is assumed that "The strength of agreement" was very good/perfect for rTGGRA6 and 7, and moderate for rTgGRA14. However, these results should be presented in the Results section in writing.
Response: The whole paragraph in lines 89-92 was removed. The same statement is already mentioned in the material and methods section (lines 260-263).
Question: The 27 rTgGRA7 positive sera were tested for the different immunoglobulin subclasses using the 3 recombinant antigens. I don't understand why 27 sera were tested for rTgGRA14 and rTgGRA6, when only 11 and 25 sera were positive for total IgG with these recombinant proteins.
Response: 27 sera tested positive using the com-ELISA kit as well as rTgGRA7. We tested the same number for rTgGRA6 and rTgGRA14 for the detection of IgG subclasses for validation of our results.
Question: How was the cut-off calculated for the IgG subclasses? Does the last paragraph in section 4.5 apply to detection of total IgG and IgG subclasses?
Response: The cut-off values for both the total IgG and IgG subclasses were calculated as the mean value at OD415nm plus three standard deviations of human sera (N = 10) previously confirmed negative using commercial ELISA and LAT kits. Yes, the last paragraph applies to both.
DISCUSSION
Lines 135 and 153: What means ICT and PVs?
Response: ICT means immunochromatographic test. Line 135 (now line 131) was revised into, “An immunochromatographic test using TgGRA7...”
PV means parasitophorous vacuole. Lines 152-153 (now lines 148) were revised into, “...is found on the parasitophorous vacuole (PV) membrane extensions linking neighboring PVs.”
Reviewer 2 Report
The manuscript "Comparative performance of recombinant GRA6, GRA7, and GRA14 for the serodetection of T. gondii infection and analysis of IgG subclasses in human sera from the Philippines" investigated the use of three recombinant proteins for the serological diagnosis of Toxoplasma gondii in humans in the Philippines.
The topic is worthy of investigation also due to public health implications.
The study is well-conducted and scientifically sounded.
Therefore, I do recommend acceptance for publication in present form in Pathogens.
Author Response
The manuscript "Comparative performance of recombinant GRA6, GRA7, and GRA14 for the serodetection of T. gondii infection and analysis of IgG subclasses in human sera from the Philippines" investigated the use of three recombinant proteins for the serological diagnosis of Toxoplasma gondii in humans in the Philippines.
The topic is worthy of investigation also due to public health implications.
The study is well-conducted and scientifically sounded.
Therefore, I do recommend acceptance for publication in present form in Pathogens.
Response: Thank you very much for your positive response.